# Genetic Diversity and Trends of Ancestral and New Inbreeding in German Sheep Breeds by Pedigree Data

**DOI:** 10.3390/ani13040623

**Published:** 2023-02-10

**Authors:** Cathrin Justinski, Jens Wilkens, Ottmar Distl

**Affiliations:** 1Institute of Animal Breeding and Genetics, University of Veterinary Medicine Hannover (Foundation), 30559 Hannover, Germany; 2vit—Vereinigte Informationssysteme Tierhaltung w.V., Heinrich-Schröder-Weg 1, 27283 Verden, Germany

**Keywords:** German sheep breeds, genetic diversity, inbreeding, pedigree analysis, ancestral inbreeding, realized effective population size

## Abstract

**Simple Summary:**

Sheep breeding is a long-standing tradition throughout Germany. Due to breeding in marginal and harsh sites, sheep developed into a large number of unique breeds adapted to many different ecosystems. In this work, we study demographic measures of genetic diversity and inbreeding trends in 35 sheep breeds using the national database of herdbook breeders in Germany. This database is a valuable resource to manage and monitor diversity in breeding populations. The loss of genetic diversity was found in all breeds studied, mainly due to genetic drift rather than unequal use of founders. The analysis of pedigree data from more than 1.4 million sheep revealed an overall measure of inbreeding of F = 0.031, an individual rate of inbreeding of ΔF_i_ = 0.0074, and a realized effective population size of *N_e_* = 91.4 with 25–75% quartiles of 0.019–0.040, 0.0040–0.0086, and 57.9–125.3, respectively. Trends in individual inbreeding were significantly positive in meat and mountain sheep, but trends in the individual rate of inbreeding were only slightly positive. Country sheep showed significantly negative trends in the rate of individual inbreeding. Ancestral inbreeding had increasing trends in all sheep breeds. Our results demonstrate the efficiency of genetic diversity management and should help to conserve endangered breeds and maintain high genetic diversity in breeds used for wool, meat, and milk production.

**Abstract:**

In Germany, many autochthonous sheep breeds have developed, adapted to mountain, heath, moorland, or other marginal sites, but breeds imported from other countries have also contributed to the domestic breeds, particularly improving wool and meat quality. Selective breeding and the intense use of rams may risk losing genetic diversity and increasing rates of inbreeding. On the other hand, breeds with a low number of founder animals and only regional popularity may not leave their endangered status, as the number of breeders interested in the breed is limited. The objective of the present study was to determine demographic measures of genetic diversity and recent as well as ancestral trends of inbreeding in all autochthonous German sheep breeds and sheep of all breeding directions, including wool, meat, and milk. We used pedigree data from 1,435,562 sheep of 35 different breeds and a reference population of 981,093 sheep, born from 2010 to 2020. The mean number of equivalent generations, founders, effective founders, effective ancestors, and effective founder genomes were 5.77, 1669, 123.2, 63.5, and 33.0, respectively. Genetic drift accounted for 69% of the loss of genetic diversity, while loss due to unequal founder contributions was 31%. The mean inbreeding coefficient, individual rate of inbreeding (∆F_i_), and realized effective population size across breeds were 0.031, 0.0074, and 91.4, respectively, with a significantly decreasing trend in ∆F_i_ in 11/35 breeds. New inbreeding, according to Kalinowski, contributed to 71.8% of individual inbreeding, but ancestral inbreeding coefficients showed an increasing trend in all breeds. In conclusion, in our study, all but one of the mountain-stone sheep breeds and the country sheep breed Wald were the most vulnerable populations, with *N_e_* < 50. The next most endangered breeds are exotic, country, and heath breeds, with average *N_e_* of 66, 83, and 89, respectively. The wool, meat, and milk breeds showed the highest genetic diversity, with average *N_e_* of 158, 120, and 111, respectively. The results of our study should help strengthen conservation program efforts for the most endangered sheep breeds and maintain a high genetic diversity in all sheep breeds.

## 1. Introduction

Sheep are one of the oldest domestic animal species in the world, and were bred long ago for their robustness, frugality, and adaptability to climatic conditions and a changing food supply. The most important breeding characteristics were milk, wool, and meat. Breeding has resulted in a very wide variety of breeds, especially in Europe, with varying degrees of adaptation to landscape and climatic conditions [1,2]. Intensive production, improvement in autochthonous breeds with other foreign and more productive breeds, and increased commercial demands, particularly since the last 50–70 years, have contributed significantly to the threats facing European sheep breeds [3].

However, due to the production system and the associated selection pressure, many of these breeding successes correlate with a decrease in the biodiversity of animal genetic resources, especially in the livestock sector, including the sheep sector [1,4]. The genetic diversity of a population is represented as a collection of alleles and genotypes. It is expressed among individuals and populations in different phenotypes, physiologies, and behaviors [5,6]. Knowledge of the pedigree can be used to monitor changes in genetic diversity under constant selection pressure [6,7]. Even if breeds are not yet endangered, it is important to analyze them when herdbook numbers are already decreasing in order to preserve remaining genetic resources, since small populations in particular harbor an increased risk of inbreeding and loss of genetic variability [8]. Maintaining genetic diversity among breeding animals in closed and small populations is very important because of the accelerated depletion of allelic distinctiveness and heterozygosity. Genetic selection and drift in small populations leads to adverse consequences, such as reduced vigor or production in animals with increased homozygosity and loss of allelic diversity [9]. For example, in a study on the inbreeding estimation in the small population of Basco-Béarnaise sheep, inbreeding depression was shown to have a significant effect on sperm motility [10].

Selective breeding is expected to reduce the fitness of animal populations through its negative effects on genetic diversity, especially in small, closed populations. The reduction in fitness caused by intensive selection for production traits is more pronounced in random mating systems due to more frequent mating between close relatives [11,12]. Today’s partly highly specialized sheep breeds are the product of a long selective breeding history and, therefore, differ breed-specifically to a greater or lesser extent in their characteristics and properties. They are grouped into economic and land breeds and can be further differentiated into meat, merino, milk, country, and hair sheep, depending on their main breeding aim or a specific objective, with only one exception. Merinos are also bred for meat production. Breeding for specific traits enforces unequal founder contributions and loss of founder genomes, leading to a reduction in genetic variability, as shown in the Swedish Gute [2], Nellore [6], Valachian [7], Xalda [13], Santa Inês [14], Zandi [15], and Afshari [16] breeds. In order to conserve the remaining animal genetic resources, they must first be documented and assessed from a population genetics perspective [1].

The development of sound genetic conservation strategies for livestock species requires, as a first step, the monitoring of existing genetic diversity within and between breeds. The availability of sufficient genetic variation in a given population provides the basis for sustained genetic improvement in economically important traits and facilitates adaptation to changing environmental conditions [17,18]. In this work, pedigree data of 35 sheep breeds bred in Germany were analyzed. Among the breeds studied, autochthonous breeds of all breeding directions (merino, meat, country, and milk) in Germany were represented. The pedigree data for our analyses were provided by vit/Verden (Vereinigte Informationssysteme Tierhaltung w.V., Verden, Germany), the information service provider for animal husbandry and breeding in Germany.

## 2. Materials and Methods

For our analysis, we used the pedigree data of 1,435,562 sheep from 35 different sheep breeds. The reference population were sheep born between 2010 and 2020 and included 981,093 animals (Table 1). Pedigree data were extracted from serv.it OviCap by Vit/Verden, Germany. The OviCap database has been managed by vit/Verden since 2007 and has been available to all sheep breeders since 2013 as an internet-based platform for viewing performance data, as well as for their own work with data from the herdbook and national evaluations of breeding values. All pedigree entries are checked for consistency during data entry and errors must be corrected before the data are accepted. Breeders can use pedigree data from OviCap to avoid inbreeding in the next generation of lambs when planning future mating. In addition, breeders can obtain information on male founders and their contribution to the animals of the current and next generation.

Estimates of demographic measures of genetic diversity were obtained for the respective reference populations of each breed using the PEDIG software [19]. Data editing and calculation of individual rates of inbreeding and realized effective population sizes were performed using SAS, version 9.4 (Statistical Analysis System, Cary, NC, USA, 2022).

In brief, we calculated the number of equivalent complete generations (GE) to determine the degree of completeness of the pedigrees in the total dataset [20] as well as the generation intervals (GI) for sires and dams [21]. The demographic measures of genetic diversity estimated included number of founders (*f*), number of effective founders (*f_e_*), effective number of founder genomes (*f_g_*), effective number of ancestors (*f_a_*), and the ratios *f_e_*/*f*, *f_a_*/*f_e_* and *f_g_*/*f_e_* [22,23] based on following formulas:$$f_{e}=\frac{1}{\sum_{k=1}^{f}q_{k}^{2}},\;f_{g}=\frac{1}{\sum_{k=1}^{f}\frac{q_{k}^{2}}{r_{k}}},\;\mathrm{and}\;f_{a}=\frac{1}{\sum_{j=1}^{a}q_{j}^{2}},$$
with *f* = number of founders, *a* = number of ancestors, *q_k_* = probability of gene origin of the individual ancestor (*k*), and *r_k_* = expected proportion of founder alleles that have been kept within the descendant population, marginal genetic contribution of an individual ancestor (*q_j_*).

In addition, we derived the amount of genetic diversity (*GD*) resulting from unequal contribution of founders and genetic drift [18,24] as follows:$$GD=1-\frac{1}{2f_{g}}$$

The amount of genetic diversity (*GD**) due to an unequal contribution of founders was:$${GD}^{*}=1-\frac{1}{2f_{e}}$$

The loss of genetic diversity resulting from unequal contribution of founders is given by 1 − *GD** and from genetic drift by *GD** − *GD*. The relative amounts of these losses of genetic diversity can be calculated as proportions of the sum of 1 − *GD** and *GD** − *GD*.

We estimated individual inbreeding coefficients (F) according to Meuwissen and Luo [25] using PEDIG [19], the genedrop method (F_gd_), ancestral inbreeding coefficients according to Ballou [26] (F_a_Bal_), ancestral (F_a_Kal_) and new (F_New_) inbreeding coefficients according to Kalinowski et al. [27], and ancestral history coefficients (AHC) defined by Baumung et al. [28] using GRAIN, version 2.2 [28,29]. In the data analysis of the reference population, we distinguished estimates for F for all animals and only for inbred animals (F_inbred_).

We calculated the degree of deviation of random mating from Hardy–Weinberg proportions as the indicator of genetic substructure by comparing the average coancestry within the parental population (Φ) with the average coefficient of inbreeding according to Meuwissen and Luo [25] in the reference population as follows [30]:$$\propto =1-\frac{1-F}{1-\Phi }$$

The individual rate of inbreeding (ΔF_i_) was adjusted to GE according to Gutiérrez et al. [31] and calculated as follows:$${\Delta F}_{i}=1-\sqrt[{GE_{i}-1}]{1-F_{i}}$$

The realized effective population size (*N_e_*) was derived from the mean ΔF_i_ ($\bar{\Delta \mathrm{Fi}}$) of the reference population [32]:$$N_{e}=\frac{1}{2\bar{{\Delta F}_{i}}}$$

The unbalanced use of male and female breeding animals is expressed by the effective number of sires (*NeffS*) and dams (*NeffD*):$$NeffS=\frac{1}{\sum_{i}s_{i}^{2}}\;and\;NeffD=\frac{1}{\sum_{i}d_{i}^{2}}$$
where *s_i_* (or *d_i_*) is the relative frequency of use of the sire or dam *i* among all sires (or dams) in the reference population [33].

The distributions of the demographic parameters *f_e_*, *f_g_, f_a_*, and their ratios, losses due to unequal contributions of founders and genetic drift, and number and effective number of dams and sires are presented for the 35 breeds using boxplots. The boxplots show, in addition to the median, the lower and upper quartiles, as well as the upper and lower whisker, each marking 1.5 times the interquartile range. In order to differentiate the 35 different breeds, we have chosen bee swarm models for the plots.

We calculated all different inbreeding coefficients’ means by breed and birth year to analyze trends in these parameters over time. The model applied included breed as a fixed effect and a linear regression coefficient within breed on birth year encoded 1 (=2010) to 11 (=2020). Pearson correlation coefficients among the different coefficients of inbreeding were calculated using SAS. In addition, we calculated Pearson correlation coefficients for individual and ancestral inbreeding coefficients between offspring and parents, as well as between both parents, using SAS.

## 3. Results

The number of animals in the pedigrees and the reference population as well as GE, GI, number of founders, effective number of founders, effective number of founder genomes, and effective number of ancestors for each of the 35 sheep breeds are shown in Table 1. The mean GE of all 35 breeds was 5.77, with 27 of the 35 breeds having a GE > 4, ranging from a minimum of 2.55 in Baraka to a maximum of 8.68 in White Polled Heath. The generation equivalent shows an increasing trend in all breeds over the birth years in the respective reference populations from 2010 to 2020. The trends are shown for each breed in Appendix A. The mean GI across all breeds was 3.83 years. Mean numbers of founders (*f*), effective number of founders (*f_e_*), effective number of founder genomes (*f_g_*), and effective number of ancestors (*f_a_*) across all breeds were 1668.9, 123.2, 33.0, and 63.5, respectively. Median values of *f_e_*, *f_g_*, and *f_a_* were 99.8, 25.6, and 53.6, respectively (Figure 1). Within the first and third quartiles were 19/35 breeds for *f_e_* (52.9–156.8), *f_g_* (18.0–42.1), and *f_a_* (36.6–76.6). The maximum number of founders was found in German Mutton Merino with 8500 and the minimum in Black Mountain with 161. The highest number of effective number of founders was in Suffolk with 477.7 and the lowest in Alpine Steinschaf with 24.1. The most effective number of founder genomes were in Suffolk with 125.6 and the lowest in Black Mountain with 10.7. The highest number of effective number of ancestors was Suffolk with a value of 225.6 and the lowest was Black Mountain with 15.3.

All ratios of *f_e_/f*, *f_a_/f_e_*, and *f_g_/f_e_*, were less than 1, which is indicative of bottlenecks, drift, and unequal use of founders in all breeds (Figure 1, Appendix A). Ratios of *f_e_/f* ranged from 0.03 (Dorper) to 0.296 (Swifter). Outside the first and third quartiles from 0.06 to 0.14 were 18/35 breeds. Ratios of *f_a_/f_e_* ranged from 0.28 (German Mutton Merino) to 0.95 (Baraka). Within first and third quartiles (0.47–0.63) were 16/35 breeds. The ratios of *f_g_/f_e_* had the lowest value in German Mutton Merino with 0.13 and the highest in Baraka with 0.60, and 17/35 breeds were within the quartiles (0.22–0.39).

In addition, we calculated how many ancestors explained 30/50/70/90/95% of the genetic diversity in the different sheep breeds (Appendix A). The mean number of ancestors explaining 30% of the gene pool of the 35 sheep breeds was 8.74. Suffolk had the highest number of ancestors with 33 explaining 30% of the gene pool, whereas Alpine Steinschaf and Black Mountain had the lowest value with only 2 ancestors. The mean number of ancestors explaining 50% of the gene pool was 23.94 on average, with ranges between 5 (SBS) and 96 (SUF). The highest number of ancestors explaining 90% of the population diversity reached 1000 ancestors in German Mutton Merino, but the lowest in Black Mountain with 48 ancestors. The average number of ancestors explaining 30%, 50%, 70%, and 90% of the population diversity across all breeds was 8.74, 23.94, 62.77, and 273.89, respectively.

The mean loss of genetic diversity due to unequal founder contributions (1 − *GD**) was 0.0066 on average across all breeds (Figure 2, Appendix A). The first and third quartile ranged from 0.0032 to 0.009 and included 17/35 breeds. Outliers were the breeds AST and SBS, with estimates of 0.0207 and 0.0196, respectively. Suffolk had the lowest estimate for the loss of genetic diversity due to unequal founder contributions with 0.001.

The mean loss of genetic diversity due to genetic drift (*GD** − *GD*) was 0.0141 and ranged from 0.0029 (SUF) to 0.036 (WGH) among all breeds. The lower quartile was at 0.0095 and the upper quartile at 0.0176, including 17/35 breeds. The loss of genetic diversity caused by genetic drift was, across breeds, on average, higher, as opposed to losses due to unequal founder contributions. Relative loss of genetic diversity resulting from genetic drift reached 69.4% on average for all breeds, whereas unequal contributions of founders were responsible for a relative loss of 30.6% of the genetic diversity (Table 2).

The different measures of inbreeding are shown in Table 3 and Figure 3, Figure 4 and Figure 5. The average F value across all breeds was 0.031, with the highest value of 0.079 (WAD) and the lowest value of 0.008 (CHA). Since the results of the inbreeding coefficients according to the genedrop method (F_gd_) are identical to F, they are not provided. Inbred animals had higher inbreeding coefficients with an overall mean of 0.051 and a range from 0.018 (MLW) to 0.100 (IDF). The proportion of inbred animals varied from 0.113 (CHA) to 0.953 (RPL), with an average of 0.660. The comparison of the degrees of parental coancestry (Φ) and average individual coefficients of inbreeding within the respective reference population for each breed yielded a range from 0.0007 (MLW) to 0.0538 (WAD), with an overall α-value of 0.0160. With exception of the breed WAD, all breeds were within the upper and lower whisker (Appendix A).

Estimates of the individual rate of inbreeding (ΔF_i_) were on average 0.0074, with a range from 0.0028 (MLW) to 0.0210 (GBS). Realized effective population size (*N_e_*) varied from 23.83 (GBS) to 179.16 (MLW), with an average of 91.35. We identified five breeds with *N_e_* values < 50, fifteen breeds with *N_e_* values of 50–100, and fifteen breeds with *N_e_* values > 100. The distribution of breeding directions in sheep breeds were significantly different when subdivided by *N_e_* values. All merino and milk sheep breeds and all but two of the seven meat sheep breeds reached *N_e_* values > 100. Most of the country sheep breeds (4/7) exhibited *N_e_* values of 50–100, two breeds (BLS, RHO) had *N_e_* values > 100 and one breed (WAD) had a *N_e_* value < 50. Among the mountain-stone sheep, breeds with *N_e_* values < 50 (AST, GBS, SBS) and *N_e_* values of 50–100 (BBS, KST, WBS) were equally represented. All but one of the four heath breeds (GGH with *N_e_* >100) had *N_e_* values of 50–100. The exotic breed NOL, had a *N_e_* value < 50, three exotic breeds had *N_e_* values of 50–100, and BDC had a *N_e_* value >100.

New inbreeding, according to Kalinowski [27], is higher than ancestral inbreeding F_a_Kal_. The mean F_a_New_ was 0.022, and its range across breeds was between 0.008 (CHA) and 0.046 (WAD). The mean F_a_Kal_ 0.009 ranged from 0 (BDC, CHA) to 0.033 (WAD). Ancestral inbreeding coefficients AHC and F_a_Bal_ were higher than classical inbreeding coefficient F. AHC, with an average of 0.09, ranged from 0.005 (CHA) to 0.261 (OUS). The mean F_a_Bal_ was 0.082, with the minimum value of 0.005 (CHA) and the maximum of 0.19 (OUS).

Trends over birth years by breed are shown for F in all and inbred animals, as well as proportion of inbred animals, in Appendix A. Appendix A presents time trends for ΔF_i_ and *N_e_* and Appendix A for F_a_Bal_, F_a_Kal_, F_a_New,_ and AHC. The linear regression coefficients on birth years within breed with their *p*-values are provided in Appendix A. Significantly positive and negative trends in F were found in 9/35 (AST, CHA, GGH, IDF, LAC, SBS, SWS, WBS, and WGH) and 4/35 (GBS, KST, OUS, and ZWS) breeds, respectively. For ΔF_i_, trends were significantly negative in 11/35 breeds (BBS, BDC, DOS, GBS, KAM, KST, NOL, OUS, SKU, WAD, and ZWS) and slightly positive but not significant in 12/35 breeds. For *N_e_*, trends were significantly negative in 1/35 breeds (CHA) and significantly positive in 2/35 breeds (BDC, ZWS). Time trends for F_a_Bal_ and AHC were positive in all breeds and the linear regression coefficient was significant in 30/35 breeds. Significantly positive trends were found in 17/35 breeds for F_a_Kal_ and significantly negative trends in 3/35 breeds, whereas for F_a_New_, significantly positive trends were found in 5/35 breeds and significantly negative trends in 7/35 breeds.

We assigned the breeds to their breeding directions and tested whether significant differences could be found in demographic measures of genetic diversity and inbreeding coefficients (Appendix A). Merino, meat, and milk breeds had the largest *f*, *f_e_*, *f_g_*, and *f_a_*, the lowest losses due to drift and unequal contributions from founders, the lowest value for α, the lowest individual rate of inbreeding, and the largest *N_e_*. In contrast, mountain-stone sheep showed the lowest *f*, *f_e_*, *f_g_*, and *f_a_*, the highest losses due to unequal contributions from founders, the highest individual rate of inbreeding, the lowest *N_e_*, and the lowest effective number of sires and dams.

On average, per breed and birth year, 1522.61 dams and 1289.25 effective number of dams were present (Figure 6). More than half of the breeds (*n* = 18/35) were within the first and third quartile of the boxplot, and three breeds were outliers (MLS, SKF, and MFS). The maximum effective number of dams was found in German Merino, German Blackhead Mutton, and German Mutton Merino with 6870.4, 4229.4, and 3949.2 animals, respectively. The lowest effective number of dams, with 83.7, was seen in the Black Mountain (Appendix A).

The numbers of sires and effective number of sires per breed and birth year ranged from 14.4 (SBS) to 263.8 (SUF) and from 6.5 (LAC) to 123.1 (TEX), respectively (Figure 7). Within the 25–75% quartiles were 17/35 breeds for the number of sires and 18/35 breeds for the effective number of sires, respectively. Outliers with a high number of sires were the breeds SUF, TEX, and SKF. The maximum effective number of sires was 123.1 for Texel, and this was the only outlier. Similar to the number of dams, the Black Mountain was one of the three breeds with the lowest effective number of sires and sires. The breed–birth–year average across breeds for effective number of sires was 41.57 (Appendix A).

Pearson correlation coefficients across breeds among the different inbreeding coefficients and proportion of inbred animals were highly positive (Table 4). The individual rate of inbreeding was strongly positively correlated with loss due to unequal use of founders. Ancestral inbreeding coefficients F_a_Bal_ and AHC had correlations with ΔF_i_ close to zero.

With a larger number of founders (*f*), effective number of founders (*f_e_*), effective number of founder genomes (*f_g_*), and effective number of ancestors (*f_a_)* per breed, inbreeding coefficients, individual rates of inbreeding, F_a_Kal_, and F_a_New_ significantly decrease (Table 5). A large effective number of sires is highly positively correlated with *f*, *f_e_*, *f_g_*, and *f_a_*. Similar correlations were obtained for effective number of dams with the exception of *f_g_*. The relative loss of genetic diversity due to drift was significantly positively correlated with *f* and *f_e_*. The decline in *f_g_* correlated more strongly with the increasing loss of drift-related diversity than with unequal contributions from founders.

## 4. Discussion

The aim of this study was to assess genetic diversity and inbreeding trends comprising all autochthonous German sheep breeds and sheep of all breeding directions including wool, meat, and milk. There is a great interest in genetic diversity studies in domestic animals, with special focus on ruminants [34]. The present study also aims to demonstrate the efficiency of breed conservation programs underway in Germany.

Completeness and depth of pedigree data is crucial for the estimation of pedigree-based measures of genetic diversity and inbreeding. All but two autochthonous and all but two imported breeds reached >3.5 equivalent generations. Equivalent generations >4 and >5 had 27/35 and 23/35 breeds, respectively. Similar to the Santa Inês sheep (2.26) [14] and the Kermani sheep (2.22) [35], the breeds Baraka (2.55), Charollais (3.08), and Lacaune (3.09) showed pedigrees with the lowest depth. Summarizing previous reports (Appendix A), equivalent generations were 5.42 on average, 24/33 with >4, and 17/33 with >5 equivalent generations, and in the present study, with an average of 5.77 equivalent generations in the same range. Because of the importance of pedigree completeness and quality, equivalent generations should be four and larger [36,37]. Six breeds (BLS, MLS, OMS, SKF, WBS, and WHH) reached equivalent generations near 8 or even >8, which is similar to the summary of previous reports on Bleu du Maine [38], Lacaune Confederation, Lacaune Ovitest [39], German Whitehead Mutton [8], Romanov in vivo [38], and Belgium Milk Sheep [40]. Breeds imported a few decades ago such as BDC, CHA, LAC, IDF, and SWS do not have deep pedigrees due to their short breeding history in Germany. NOL was developed in recent years as hair sheep in Germany and has, therefore, only a short history as its own officially recognized breed. The mountain sheep GBS and SBS are color variants arisen from WBS and have been recognized as independent breeds with their own herdbook only in the last few decades.

In agreement with previous reports (Appendix A), ratios of *f_e_/f*, *f_g_/f_e_*, and *f_a_/f_e_* < 1 indicate the loss of genetic diversity since the founder generation. All breeds had ratios <1, and *f_e_/f* was, on average, the smallest ratio in all breeds, while *f_a_/f_e_* was the largest. A similar distribution was also found in French sheep breeds [38], Afshari [16], Nellore [6], Iran-Black [41], and Iranian Baluchi [42], but not in Zandi [15], Kermani [35], Iranian Lori-Bakhtiari [43], and Swedish Gute [2]. The unequal use of founders and the presence of genetic drift, indicated by *f_e_/f* <1, was more obvious in merino (0.042) and heath sheep (0.069) than in milk (0.175), exotic (0.166), and mountain-stone (0.123) sheep breeds. In exotic and mountain-stone sheep, *f_e_/f* may be related with shorter historical pedigree records and, therefore, overestimated [44]. The lowest *f_e_/f* estimates in previous reports were 0.09 in Roussin de la Hague, Charmoise [38], Bharat Merino [45], and Iran-Black [41]. In contrast to previous reports, we found *f_e_/f* <0.09 in 17/35 breeds. One possible explanation for this result is that in the OviCap database, important founders could be traced back over 15 generations, in some cases even more. Relative loss of genetic diversity due to drift was much more important in all breeds, with the exception of GBS and NOL. In these breeds, losses due to unequal founder contributions were larger compared to losses due to genetic drift.

The merino breed MFS, the heath breed GGH, and the milk breed OMS showed the largest drift-related losses in our study, with *f_g_/f_e_* ratios of 0.131, 0.165, and 0.166, respectively. The French breeds Roussin de la Hague (0.14), Romanov (0.15), and Charmoise (0.17) had similarly low *f_g_/f_e_* ratios and, thus, high losses of founder genomes [38]. Severe bottlenecks may be assumed in 9/35 breeds with *f_a_/f_e_* below the first quartile (0.472). All breeding directions were represented among these breeds with *f_a_/f_e_* <0.472. The merino breed MFS had the lowest *f_a_/f_e_* ratio of 0.281, followed by the exotic breed ZWS (0.381) and the heath breed GGH (0.383). In Valachian sheep, an even lower *f_a_/f_e_* ratio of 0.25 [7] was reported, while German Whitehead Mutton [8] and Charmoise [38] reached estimates of 0.36 and 0.37.

If there is evidence of genetic bottlenecks, as was the case in our study for all breeds studied, the effective number of sires and dams should be reviewed. The numbers of effective sires and dams estimated for each breed and birth year strongly correlated with the number of founders, effective number of founders, founder genomes and ancestors, and the ratios *f_e_/f*, *f_g_/f_e_*, and *f_a_/f_e_*, as well as the individual rate of inbreeding and realized effective population size. Particularly, breeds with small population sizes have to control the number of progeny per sire in order to avoid popular sire effects with negative consequences of an increase in individual rate of inbreeding and decrease in realized effective population size. De Vries showed with a simulation study that it is essential to have a low ram-to-ewe ratio to maintain genetic diversity [46]. Furthermore, most of the rams used for breeding are still young, about 2 years old, and are replaced only to a very limited extent. Windig investigated different mating schemes for this purpose and proved that in all of the different schemes, ΔF_i_ decreased and, thus, inbreeding rates could be restricted by means of targeted mating [47]. Ghafouri-Kesbi also recommends a scheme to keep the inbreeding rate low with targeted mating and further points out the importance of sufficiently high generation intervals [16].

We found no evidence of line breeding or a significant increase in inbreeding in offspring of already inbred parents, as correlations of individual and ancestral inbreeding measures between parents were very close to zero for most breeds (Appendix A). In addition, the correlations of F_New_ of the animal with the measures of ancestral inbreeding between the dam and sire were close to zero, indicating that new inbreeding in the offspring was not associated with ancestral inbreeding in the parents.

The negative trends in effective number of sires for merinos, meat, and heath breeds were due to decreasing population size across birth years, with the exception of MLW. For this breed, decreasing population size was not responsible for the decreasing effective number of sires with birth years. For the exotic breeds, the positive trends are related with increasing population size, but for the mountain-stone and country sheep, the positive trends are equally associated with improved management of breed diversity and increasing population size. For COF, the main factor appears to be improved breed management. This was demonstrated by a model extended by a linear regression of population size per birth year within breed.

The realized effective population size plays an important role in breed conservation and assessing the endangerment status of breeds (Appendix A). We derived *N_e_* from the individual rate of inbreeding (ΔF_i_) to obtain estimates independent of the number of generations recorded in the pedigree data. Both parameters varied in a wide range, similar to previous studies, and showed similar distributions (Appendix A).

In the present study, the parameters most strongly correlated with ΔF_i_ across breeds were *f_a_/f_e_* (0.623), *f_g_/f_e_* (0.667), *f_e_* (−0.531), and loss of genetic diversity due to unequal contributions of founders (0.799). New inbreeding (F_a_New_) had a higher impact on ΔF_i_ (0.755) and *N_e_* (−0.756) than the classical inbreeding coefficient (0.649, −0.699), because F_a_New_ had a higher relative contribution to inbreeding. Inverse relationships across breeds were found between *N_e_* and *f_a_/f_e_* (−0.590), *f_g_/f_e_* (−0.529), *f_e_* (0.652), *f* (0.687), and unequal founder contributions (−0.716). In previous reports, we found strong correlations between ΔF_i_ and *f_e_/f_a_* (0.873), *f_g_/f_e_* (0.747), and *f_g_/f_a_* (0.632), and between *N_e_* and *f_g_/f_e_* (−0.668) and *f_g_/f_a_* (−0.549). In the across-breed comparisons of the present study, loss of genetic diversity due to unequal contribution of founders was found to be the most important factor leading to increasing individual rates of inbreeding and smaller realized effective population sizes. The greatest influence on the loss of genetic diversity due to unequal contributions from founders was the effective number of sires and dams. Data from the across-breed comparisons showed that high effective numbers of dams and high numbers of founders, and high effective numbers of sires and high effective numbers of founders were mutually dependent. Our data suggest that breeding organizations and breeders should be concerned with keeping the number of breeding dams and sires high and using dams and sires as equally as possible in breeding programs.

Breeds with *N_e_* values in the critical ranges of 50–100 are considered endangered. In the present study, 15/35 breeds and 5/35 breeds were below a *N_e_* of 100 and a *N_e_* of 50, respectively. The latter five breeds would reach an estimated increase in inbreeding of 25.89% (GBS), 23.06% (SBS), 19.10% (AST), 17.11% (WAD), and 15.95% (NOL) in 50 years and, thus, be above the critical threshold of a 5% increase in inbreeding in 50 years [48]. The trends in ΔF_i_ were significantly decreasing in GBS, NOL, and WAD and, therefore, we may expect a lower increase in inbreeding rates for these breeds than calculated from the present breed averages. In the breed AST, a slightly negative trend for rate of inbreeding was observed, but for the breed SBS, a slightly increasing trend was observed, which may lead to a lower-than-expected increase in inbreeding in AST, but a higher-than-expected increase in inbreeding in SBS. Only 7/35 (MLS, MFS, MLW, SKF, TEX, RHO, and GGH) and 24/35 breeds are expected not to miss the threshold of 5% and 10% increase in inbreeding in 50 years.

The breeds classified as threatened according to *N_e_* < 100 can be grouped into imported and autochthonous breeds (Appendix A). While breeds such as Ile-de-France, Dorper, and Zwartbles were imported from France, South Africa, and The Netherlands for meat production, autochthonous breeds including country (BRI, COF, LES, RPL, and WAD), all mountain-stone (e.g., AST, GBS, and SBS), and heath breeds (SKU, WGH, and WHH) were bred in marginal sites under harsh climatic conditions. Since these breeds are adapted to these specific ecosystems, they gained little to no recognition in other regions. In addition, these breeds had mostly fewer foundation animals. Despite these limitations, these autochthonous breeds can be conserved through appropriate breeding management, which is supported via the OviCap database. In addition, the use of marginal landscapes in breeding sheep will increase in the future to save natural areas, and this could create demand for sheep suitable for marginal and harsh ecosystems.

Negative effects of inbreeding are assumed to be mainly caused by recent inbreeding than by ancestral inbreeding, as the frequency of deleterious alleles is expected to decrease over the time through selection [29]. Trends in ancestral inbreeding were positive in all breeds, but on average across all breeds, new inbreeding was attributable to 76.9% of individual inbreeding. Among the breeds with *N_e_* < 50, F_a_New_ is responsible for 72.7% of individual inbreeding on average, while in the mountain-stone breed AST and country breed WAD, which belong to breeds with *N_e_* < 50, F_a_New_ is only responsible for less than 60% of individual inbreeding. In the breeds with *N_e_* > 100, the proportion of F_a_New_ to individual inbreeding reaches 85.9% on average. In summary, the ancestral and new inbreeding coefficients by Kalinowski [27] differ significantly among breeds and breeding directions, and, therefore, may have different impacts on the future development of ΔF_i_ and *N_e_*. In particular, in breeds with *N_e_* > 100, F_a_New_ has to be regarded in planning future mating, as well as loss of genetic diversity due to unequal contributions of founders and effective number of dams and sires. Among breeding directions, heath sheep had the lowest proportion of F_a_New_ in individual inbreeding, while milk sheep had the highest proportion of F_a_New_ in individual inbreeding. As the proportion of F_a_New_ in individual inbreeding decreases, the loss of genetic diversity due to drift becomes more important for maintaining genetic variability, and, thus, the effective number of founder genomes.

## 5. Conclusions

The present study shows the development of demographic measures of genetic diversity and trends of inbreeding for autochthonous sheep breeds and breeds for wool, meat, and milk production in Germany. The across-breed analysis revealed losses of genetic diversity, mainly due to unequal contributions of founders in the past. This parameter had the largest impact in all 35 breeds on new inbreeding (F_a_New_), and new inbreeding was much more important for individual rates of inbreeding and realized effective population sizes compared to ancestral inbreeding and the classical inbreeding coefficient. Trends for individual rates of inbreeding were negative in 11 of the 35 breeds analyzed, which may demonstrate the efficiency of maintaining breed diversity. However, the large differences in population structure and breeding history in the different breeds makes it necessary to consider the trends for new and ancestral inbreeding and their impact on breed conservation. In addition, we demonstrated significant differences between the different breeding directions. Mountain-stone sheep breeds had the lowest average *Ne* value with <50, the lowest number of founders and founder genomes, a relatively high coancestry coefficient on average, positive trends for ancestral inbreeding (F_a_Kal_), but negative trends for new inbreeding (F_a_New_) and an increasing trend in population sizes and number of effective sires. On the contrary, merino breeds had an average *N_e_* > 150 and an expected increase in inbreeding <5% within 50 years, but a negative trend for population sizes and number of effective sires. Annual analyses of demographic data and their trends can be made available through the OviCap national database. This should help to critically review genetic diversity conservation efforts and focus on the most endangered breeds. Along with these analyses, breed associations and breeders can be informed on unequal use of founders and which founder lines are highly threatened and need special consideration in the breeding program. The results of the present study should strengthen the efforts for maintaining the genetic diversity of sheep breeds with a focus on the different ecosystems in Germany.

## Figures and Tables

**Figure 1 animals-13-00623-f001:**
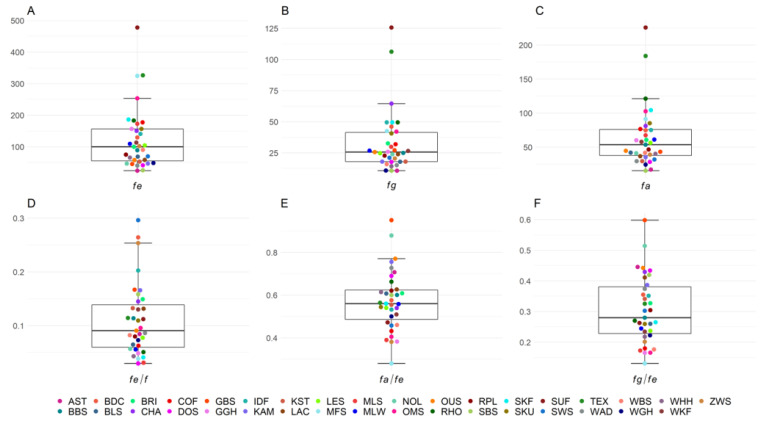
Boxplots of genealogical estimators of genetic diversity for 35 sheep breeds in Germany with (**A**) effective number of founders (*f_e_*), (**B**) effective number of founder genomes (*f_g_*), (**C**) effective number of ancestors (*f_a_*) and their ratios, and (**D**–**F**) representing possible bottlenecks in the population.

**Figure 2 animals-13-00623-f002:**
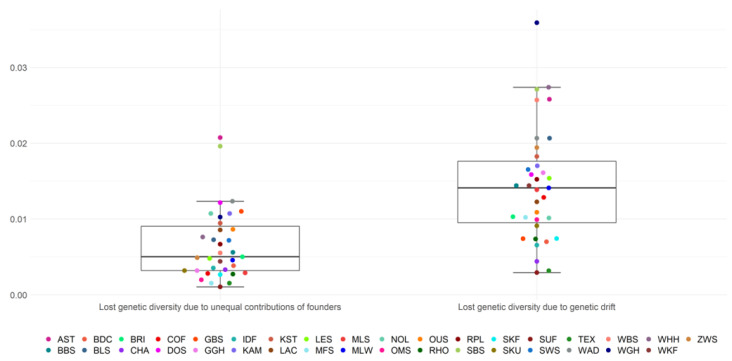
Loss of genetic diversity from unequal contributions of founders and genetic drift.

**Figure 3 animals-13-00623-f003:**
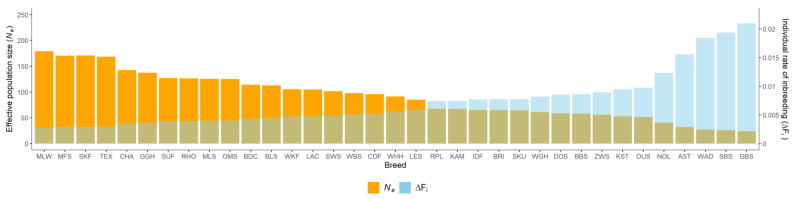
Individual rate of inbreeding (ΔF_i_) and realized effective population size (*N_e_*) in 35 sheep breeds.

**Figure 4 animals-13-00623-f004:**
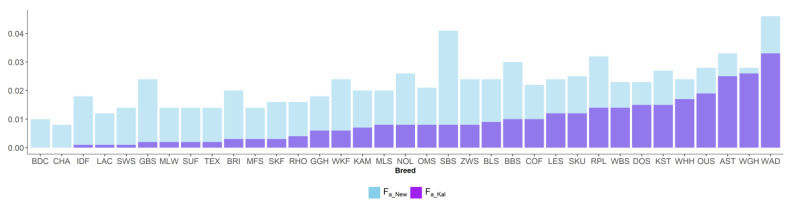
Inbreeding coefficients of Kalinowski ancestral (F_a_Kal_) and new (F_New_) in 35 sheep breeds.

**Figure 5 animals-13-00623-f005:**
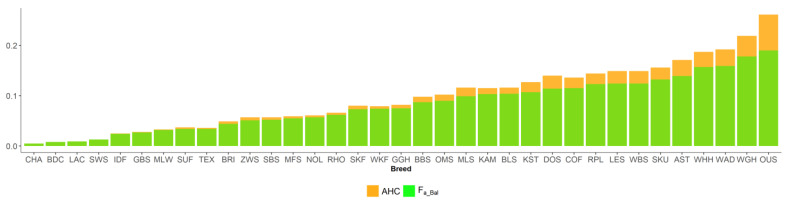
Ancestral inbreeding coefficients in 35 sheep breeds, according to Ballou (F_a_Bal_) and Baumung (AHC).

**Figure 6 animals-13-00623-f006:**
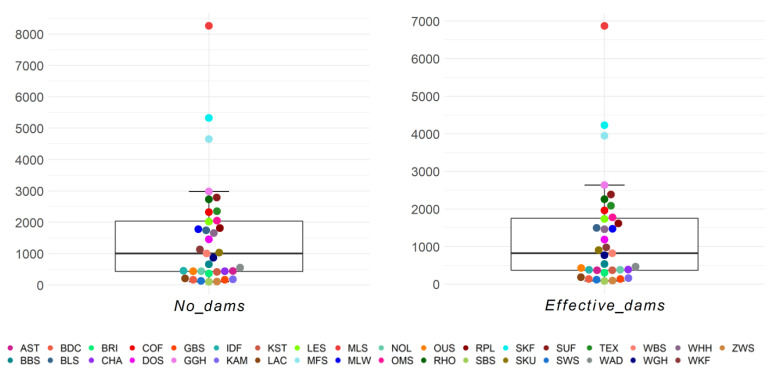
Number of dams (No_dams) and effective number of dams (Effective_dams) as birth year averages for 35 sheep breeds.

**Figure 7 animals-13-00623-f007:**
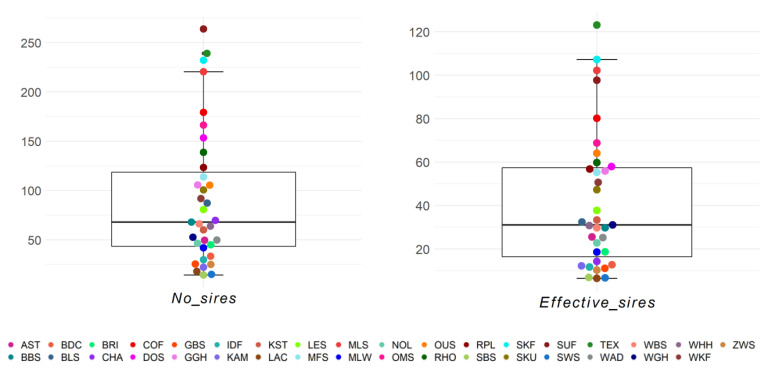
Number of sires (No_sires) and effective number of sires (Effective_sires) as birth year averages for 35 sheep breeds.

**Table 1 animals-13-00623-t001:** Pedigree analysis of 35 sheep breeds in Germany, their breeding direction (BD), number of sheep in the complete pedigree file (N_Ped_), reference population (N_Ref_), number of equivalent generations (GE), generation interval (GI), number of founders (*f*), effective number of founders (*f_e_*), effective number of founder genomes (*f_g_*), and effective number of ancestors (*f_a_*).

Code	BD	Sheep Breed	N_Ped_	N_Ref_	GE	GI	*f*	*f_e_*	*f_g_*	*f_a_*
AST	MON	Alpine Steinschaf	10,420	8729	4.71	3.70	286	24.1	10.7	17.0
BBS	MON	Brown Mountain	22,961	13,510	5.79	4.06	784	89.2	25.0	53.6
BDC	EXO	Berrichon du Cher	4680	3101	3.51	3.45	491	129.7	46.1	74.7
BLS	CON	Bentheim	46,173	29,973	7.83	3.76	1064	68.8	17.9	41.8
BRI	CON	Carinthian	10,669	6386	4.07	3.79	669	99.8	32.7	60.7
CHA	MEA	Charollais	11,237	8044	3.08	3.21	1039	150.5	64.6	81.2
COF	CON	Coburg	70,156	43,072	7.30	3.82	2828	177.4	31.9	76.6
DOS	MEA	Dorper	36,057	28,347	5.72	3.43	1380	41.1	17.8	28.4
GBS	MON	Baraka	3938	3334	2.55	3.54	272	45.4	27.2	43.1
GGH	HEA	German Grey Heath	69,369	41,982	7.39	3.92	3226	156.8	25.9	60.0
IDF	MEA	Ile-de-France	14,021	9066	3.77	3.94	696	140.9	49.5	75.0
KAM	EXO	Kamerun	7404	3313	4.59	3.99	281	46.6	18.0	35.2
KST	MON	Krainer Steinschaf	9671	7234	5.41	3.87	399	52.9	18.1	29.4
LAC	MIL	Lacaune	4652	3632	3.09	3.17	444	58.4	24.0	36.6
LES	CON	Leine	42,949	37,108	7.44	3.83	1348	104.4	24.8	56.4
MFS	MER	German Mutton Merino	132,413	85020	6.20	4.07	8500	324.5	42.5	91.1
MLS	MER	German Merino	204,494	172,172	8.30	4.02	5546	172.8	29.9	67.4
MLW	MER	Merino Longwool	61,216	30,546	6.42	3.82	1959	109.3	26.8	61.0
NOL	EXO	Nolana	11,920	8073	3.68	3.57	821	46.6	24.0	40.9
OMS	MIL	East Friesian	71,159	42,427	7.98	3.08	2644	253.3	42.1	102.6
OUS	EXO	Ouessant	10,051	4927	6.43	3.67	638	57.9	25.6	44.6
RHO	CON	Rhön	78,095	45,228	6.20	4.20	3618	183.2	49.5	121.3
RPL	CON	Pomeranian Coarsewool	56,965	31,288	6.78	4.43	940	74.9	22.8	46.5
SBS	MON	Black Mountain	3903	2110	3.42	3.73	161	25.5	10.7	15.3
SKF	MEA	German Blackhead Mutton	128,839	104,178	7.71	3.92	4554	186.4	49.5	104.4
SKU	HEA	Skudde	32,747	16,496	6.03	4.34	1431	156.6	40.7	85.2
SUF	MEA	Suffolk	68,136	54,635	5.17	3.54	4260	477.7	125.6	225.6
SWS	MEA	Swifter	4608	3309	3.78	3.03	235	69.6	21.1	31.9
TEX	MEA	Texel	58,223	45,535	5.76	6.42	2860	26.3	106.2	184.0
WAD	CON	Wald	17,172	9801	5.49	4.96	470	40.5	15.2	29.4
WBS	MON	White Mountain	30,188	19,491	8.05	3.77	1103	90.6	16.0	41.7
WGH	HEA	German White Heath	18,158	12,160	7.42	3.74	667	48.7	10.8	24.4
WHH	HEA	White Polled Heath	41,306	23,289	8.68	3.85	1525	65.5	14.3	40.2
WKF	MEA	German Whitehead Mutton	38,390	21,464	7.52	3.60	871	113.4	26.6	57.8
ZWS	EXO	Zwartbles	3222	2113	4.52	3.01	402	102	20.6	38.9

Abbreviations for breeding directions: country: CON, exotic: EXO, heath: HEA, meat: MEA, merino: MER, milk: MIL, mountain-stone: MON.

**Table 2 animals-13-00623-t002:** Relative loss of genetic diversity due to unequal contributions of founders (Loss_founder_) and genetic drift (Loss_drift_) in %.

Breed	Loss_founder_ (%)	Loss_drift_ (%)
AST	44.56	55.44
BBS	27.99	72.01
BDC	35.51	64.49
BLS	26.02	73.98
BRI	32.73	67.27
CHA	42.92	57.08
COF	17.99	82.01
DOS	43.41	56.59
GBS	59.80	40.20
GGH	16.52	83.48
IDF	35.15	64.85
KAM	38.67	61.33
KST	34.12	65.88
LAC	41.11	58.89
LES	23.74	76.26
MFS	13.11	86.89
MLS	17.27	82.73
MLW	24.49	75.51
NOL	51.42	48.58
OMS	16.61	83.39
OUS	44.25	55.75
RHO	27.04	72.96
RPL	30.47	69.53
SBS	41.96	58.04
SKF	26.53	73.47
SKU	25.96	74.04
SUF	26.29	73.71
SWS	30.29	69.71
TEX	32.55	67.45
WAD	37.41	62.59
WBS	17.67	82.33
WGH	22.24	77.76
WHH	21.80	78.20
WKF	23.44	76.56
ZWS	20.15	79.85

**Table 3 animals-13-00623-t003:** Coefficients of inbreeding according to Meuwissen and Luo for all animals (F) and inbred animals (F_inbred_), proportion of inbred individuals (Inbred), average coancestry within the parental population (Φ), and degree of deviation (α) of random mating from Hardy–Weinberg proportions in the reference populations of birth years 2010 to 2020 of the 35 sheep breeds.

Breed	F	F_inbred_	I_nbred_	Φ	α
AST	0.058	0.073	0.792	0.139	0.025
BBS	0.040	0.051	0.791	0.087	0.010
BDC	0.011	0.048	0.217	0.008	0.000
BLS	0.030	0.034	0.880	0.104	0.009
BRI	0.024	0.045	0.533	0.044	0.003
CHA	0.008	0.070	0.113	0.005	0.000
COF	0.032	0.041	0.786	0.115	0.010
DOS	0.038	0.052	0.736	0.114	0.015
GBS	0.027	0.087	0.305	0.027	0.002
GGH	0.024	0.031	0.771	0.075	0.006
IDF	0.020	0.100	0.196	0.024	0.001
KAM	0.027	0.045	0.599	0.103	0.007
KST	0.041	0.059	0.698	0.107	0.015
LAC	0.013	0.056	0.228	0.009	0.001
LES	0.036	0.042	0.860	0.124	0.012
MFS	0.017	0.027	0.614	0.055	0.003
MLS	0.029	0.031	0.921	0.099	0.008
MLW	0.016	0.018	0.854	0.032	0.002
NOL	0.035	0.062	0.561	0.057	0.008
OMS	0.029	0.036	0.795	0.090	0.008
OUS	0.047	0.058	0.809	0.190	0.019
RHO	0.020	0.026	0.767	0.062	0.004
RPL	0.041	0.043	0.925	0.123	0.014
SBS	0.049	0.096	0.510	0.052	0.008
SKF	0.019	0.020	0.932	0.073	0.003
SKU	0.037	0.048	0.765	0.132	0.012
SUF	0.016	0.029	0.535	0.034	0.002
SWS	0.015	0.052	0.276	0.013	0.001
TEX	0.014	0.021	0.645	0.034	0.002
WAD	0.079	0.097	0.819	0.159	0.033
WBS	0.037	0.046	0.803	0.124	0.014
WGH	0.054	0.065	0.826	0.178	0.026
WHH	0.042	0.050	0.834	0.157	0.017
WKF	0.030	0.033	0.910	0.074	0.006
ZWS	0.032	0.078	0.408	0.051	0.008

**Table 4 animals-13-00623-t004:** Correlation coefficients across breeds between inbreeding coefficients for all (F), inbred animals (F_inbred_), proportion of inbred animals (Inbred), individual rate of inbreeding (${\Delta F}_{i}$), realized effective population size (*N_e_*), effective number of sires (*NeffS*), effective number of dams (*NeffD*), and inbreeding coefficients according to Ballou (F_a_Bal_), Kalinowski (F_a_Kal_, F_a_New_), and Baumung (AHC), losses due to genetic drift (Loss_drift_), and unequal founder contributions (Loss_founder_). *p*-values < 0.05 are marked (*).

Parameter	F	F_a_Bal_	F_a_Kal_	F_a_New_	AHC	Loss_drift_	Loss_founder_
F	-	0.795 *	0.943 *	0.941 *	0.790 *	0.676 *	0.645 *
Inbred	0.492 *	0.733 *	0.544 *	0.424 *	0.697 *	0.366 *	−0.025
F_inbred_	0.496 *	0.735 *	0.548 *	0.425 *	0.700 *	0.374 *	−0.020
${\Delta F}_{i}$	0.649 *	0.181	0.440 *	0.755 *	0.192	0.287	0.799 *
*N_e_*	−0.699 *	−0.367 *	−0.539 *	−0.756 *	−0.379 *	−0.401 *	−0.716 *
*NeffS*	−0.144	0.151	−0.059	−0.177	0.144	−0.367 *	−0.515 *
*NeffD*	−0.213	0.070	−0.139	−0.254	0.044	−0.189	−0.490 *
Loss_drift_	-	0.614 *	0.694 *	0.580 *	0.594 *	-	-
Loss_founder_	-	0.290	0.525 *	0.681 *	0.300	0.600*	-

**Table 5 animals-13-00623-t005:** Correlation coefficients across breeds between inbreeding coefficients for all (F), inbred animals (F_inbred_), proportion of inbred animals (Inbred), individual rate of inbreeding (${\Delta F}_{i}$), realized effective population size (*N_e_*), inbreeding coefficients according to Ballou (F_a_Bal_), Kalinowski (F_a_Kal_, F_a_New_) and Baumung (AHC), relative (%) and absolute losses due to genetic drift (Loss_drift_), and unequal founder contributions (Loss_founder_), effective number of sires (*NeffS*) and dams (*NeffD*) with number of founders (*f*), effective number of founders (*f_e_*), effective number of founder genomes (*f_g_*), and effective number of ancestors (*f_a_*). *p*-values < 0.05 are marked (*).

Parameter	*f*	*f_e_*	*f_g_*	*f_a_*
F	−0.349 *	−0.513 *	−0.560 *	−0.523 *
Inbred	0.271	−0.012	−0.238	−0.030
F_inbred_	0.268	−0.021	−0.249	−0.041
${\Delta F}_{i}$	−0.500 *	−0.531 *	−0.393 *	−0.480 *
*N_e_*	0.687 *	0.652 *	0.519 *	0.595 *
F_a_Bal_	−0.092	−0.327	−0.461 *	−0.349 *
F_a_Kal_	−0.268	−0.439 *	−0.501 *	−0.456 *
F_a_New_	−0.389 *	−0.521 *	−0.543 *	−0.518 *
AHC	−0.112	−0.333	−0.449 *	−0.350 *
Loss_drift (%)_	0.553 *	0.416 *	0.038	0.229
Loss_founder (%)_	−0.553 *	−0.416 *	−0.038	−0.229
Loss_drift_	−0.331 *	−0.595 *	−0.726 *	−0.682 *
Loss_founder_	−0.567 *	−0.702 *	−0.581 *	−0.667 *
*NeffS*	0.682 *	0.697 *	0.583 *	0.705 *
*NeffD*	0.870 *	0.536 *	0.273	0.419 *

## Data Availability

Restrictions apply to the availability of these data. Data were obtained from vit/Verden and are available on reasonable request from the authors with the permission of the German sheep breeding associations.

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
