# Peer review of "Genetic Diversity and Trends of Ancestral and New Inbreeding in German Sheep Breeds by Pedigree Data"

_animals, 2023, doi:10.3390/ani13040623_

Round 1

Reviewer 1 Report

I appreciate the efforts put by authors for analyzing such a large database for 35 breeds that actually represent the whole country. The methods used  are already known and standard. I have a few queries as below:

1.     what was the reason to use PEDIG? Instead of ENDOG?

2.     What was the accuracy of pedigree data? Have you accounted for recording errors, If this data belonged to field?

3.     Line 142; Fa: replace with fa

4.     Line 162: what is meant by whole reference population in your data? Did you categorise your data in complete versus a reference population?

5.     L 214: abrupt starting. Boxplots for what?

6.     Methods section is too large. Most of the readers will already be knowing about common concepts that you are discussing at length in the methods section. Please show brevity for this section.

7.     L227: what is 27/35?

8.     You have used several methods for estimation of inbreeding. Which, according to you is best and why?

9.     Table5: it is known that with large number of founders you will have low inbreeding. What new inference could you present from this analysis in Table 5?

10.  L485-86: rewrite. Not clear.

11.  L529-530: why so? New inbreeding had higher impact?

12.  L599-600: rewrite properly.

13.  Paper is too large and therefore it is suggested that authors may reduce it to present significant results.

14.  I do not see any recommendation here. This was a demographic analysis and reader may expect something out of it.

Author Response

We thank the reviewer for the valuable comments and recommendations to improve our manuscript. We revised our manuscript accordingly.

Reviewer 1:

  1. what was the reason to use PEDIG? Instead of ENDOG?

Response:

Endog is very popular program with the most options for such calculations.

We did our calculations on a linux server. Due to the large dataset, calculations on a PC would have taken too much time. On the linux server, we are more flexible with supporting programs for calculations with more than just one breed.  

  1. What was the accuracy of pedigree data? Have you accounted for recording errors, If this data belonged to field?

In sheep, electronic ear tags are used which reduces errors in recording. All pedigree data are checked by vit/Verden for errors and data must be corrected. In addition, sheep are registered in the HIT database, which is mandatory according to EU legislations. In this way, data integrity and correctness is ensured. Testing for scrapie resistence genotypes in Germany since more than 15 years improves accuracy of pedigrees, particularly in sires. In addition, we checked pedigree data for correctness and could not detect errors. PEDIG software also checks for pedigree errors.

Amended:

Lines 231-241

The OviCap database has been managed by vit/Verden since 2007 and has been available to all sheep breeders since 2013 as an internet-based platform for viewing performance data, as well as for their own work with data from the herdbook and national evaluations of breeding values. All pedigree entries are checked for consistency during data entry and errors must be corrected before the data are accepted. Breeders can use pedigree data from OviCap to avoid inbreeding in the next generation of lambs when planning future matings. In addition, breeders can obtain information on male founders and their contribution to the animals of the current generation.

  1. Line 142; Fa: replace with fa

Amended: fa

Text completey revised.

Lines 242-256 and 257-302:

In brief, we calculated the number of equivalent complete generations (GE) to determine the degree of completeness of the pedigrees in the total dataset [20] as well as the generation intervals (GI) for sires and dams [21]. The demographic measures of genetic diversity estimated included number of founders (f), number of effective founders (fe), effective number of founder genomes (fg), effective number of ancestors (fa) and the ratios fe/f, fa/fe and fg/fe [22,23] based on following formulas:

 , , and ,

with f = number of founders, a = number of ancestors, qk = probability of gene origin of the individual ancestor (k), rk = expected proportion of founder alleles that have been kept within the descendant population, marginal genetic contribution of an individual ancestor (qj).

  1. Line 162: what is meant by whole reference population in your data? Did you categorise your data in complete versus a reference population?

Amended:

Lines 269-271:

In the data analysis of the reference population, we distuinguished estimates for F for all animals and only for inbred animals (Finbred).

  1. L 214: abrupt starting. Boxplots for what?

Amended:

Lines 290-295:

The distributions of the demographic parameters fe, fg, fa, and their ratios, losses due to genetic drift and unequal contribution of founders, number and effective number of dams and sires are presented for the 35 breeds using boxplots. The boxplots show, in addition to the median, the lower and upper quartiles, as well as the upper and lower whisker, each marking 1.5 times the interquartile range. In order to differentiate the 35 different breeds, we have chosen bee swarm models for the plots.

  1. Methods section is too large. Most of the readers will already be knowing about common concepts that you are discussing at length in the methods section. Please show brevity for this section.

Amended: this paragraph has been shorted and completely revised.

Lines 242-302: revised methods. See also above.

  1. L227: what is 27/35?

Amended:  

Line 309: 27 of 35 breeds

  1. You have used several methods for estimation of inbreeding. Which, according to you is best and why?

Comment: this is dependent on the population you are analyzing. In our case, new inbreeding turned out to have the largest impact in realized effective population size. Thus, new inbreeding seems to be most important. But, as we outlined, you should look at various measeures to see their relationships in order to make the best recommendation for breeding. We have summarized this in our discussion and conclusion. See below.

Amended:

Lines 720-729:

The across-breed analysis revealed losses of genetic diversity mainly due to unequal contributions of founders in the past. This parameter had the largest impact in all 35 breeds on new inbreeding (Fa_New), and new inbreeding was much more important for individual rates of inbreeding and realized effective population sizes compared to ancestral inbreeding and the classical inbreeding coefficient. Trends for individual rates of inbreeding were negative in 11 of the 35 breeds analysed, which may demonstrate the efficiency of maintaining breed diversity.

  1. Table5: it is known that with large number of founders you will have low inbreeding. What new inference could you present from this analysis in Table 5?

Amended:

Lines 643-653:

In the across-breed comparisons of the present study, loss of genetic diversity due to unequal contribution of founders was found to be the most important factor leading to increasing individual rates of inbreeding and smaller realized effective population sizes. The greatest influence on the loss of genetic diversity due to unequal contributions from founders was the effective number of sires and dams. Data from the across-breed comparisons showed that high effective numbers of dams and high numbers of founders, and high effective numbers of sires and high effective numbers of founders were mutually dependent. Our data suggest that breeding organizations and breeders should be concerned with keeping the number of breeding dams and sires high and using dams and sires as equally as possible in breeding programs.

  1. L485-86: rewrite. Not clear.

Amended:

Sentence deleted. Results removed from discussion.

  1. L529-530: why so? New inbreeding had higher impact?

Amended:

Lines 636-638:

New inbreeding (Fa_New) had an higher impact on ΔFi (0.755) and Ne (-0.756) than the classical inbreeding coefficient (0.649, -0.699), because Fa_New had a higher relative contribution to inbreeding.

Lines 643-653:

In the across-breed comparisons of the present study, loss of genetic diversity due to unequal contribution of founders was found to be the most important factor leading to increasing individual rates of inbreeding and smaller realized effective population sizes. The greatest influence on the loss of genetic diversity due to unequal contributions from founders was the effective number of sires and dams. Data from the across-breed comparisons showed that high effective numbers of dams and high numbers of founders, and high effective numbers of sires and high effective numbers of founders were mutually dependent. Our data suggest that breeding organizations and breeders should be concerned with keeping the number of breeding dams and sires high and using dams and sires as equally as possible in breeding programs.

  1. L599-600: rewrite properly.

Amended:

Lines 724-726:

Trends for individual rates of inbreeding were negative in 11 of the 35 breeds analysed, which may demonstrate the efficiency of maintaining breed diversity.

  1. Paper is too large and therefore it is suggested that authors may reduce it to present significant results.

Amended: We reduced many parts of the manuscript.

  1. I do not see any recommendation here. This was a demographic analysis and reader may expect something out of it.

Amended:

Lines 739-744:

Annual analyses of demographic data and their trends can be made available through the OviCap national database. This should help to critically review genetic diversity conservation efforts and focus on the most endangered breeds. Along with these analyses, breed associations and breeders can be informed on unequal use of founders and which founder lines are highly threatened and need special consideration in the breeding program.

Reviewer 2 Report

The paper describes different demographic measures of genetic diversity based on pedigree information to analyze the current scenario on German sheep breeds.  I believe that the topic is interesting and the paper is well structured and written.

Any way I have some comments, in detail:

Abstrac.

L45. It would be worth to mention the Ne range of ‘the next most endangered breeds’

L46. Perhaps a sort description of the obtained genetic diversity on other breeds groups (e.i. land sheep, country sheep,…)

Introduction.

L60. Among the treats affecting European breeds, I would mention also the introduction of other foreign and more productive breeds that endangered autochthonous breeds.

L63. When the ‘livestock sector’ is mentioned, the whole sector is considered or the sheep sector?

L84. Please clarify if the described categories depending on breeding directions are fixed, there is no double purpose breeds?

L91. May there be a typo, it seem like it should not be a full stop.

L101. I would rewrite the sentence as ‘autochthonous breeds of all breeding directions of German are represented’ to clarify that the considered breeding directions correspond to the autochthonous breeds.

M&M.

This point contains a lot of measures being long and messy, what makes difficult the understanding. Please consider adding subtitles or any other organizer to clarify and ease.

L107. Please provide a table showing the number of individuals in each pedigree, reference population and any other interesting descriptive data. Also it would be helpful for the general understanding to specify what the breeding direction of each breed is. Some of these things are included in Table 1, but it is not specified here and the information gets lost over the work.

L126-L132. Please make sure that all the elements included into the formulas are described into the text. As far as I can see ‘a’, or ‘j’ are not described in some points.

L135. I would specify which is the ‘individual contribution of founder’ that has to be replaced into the formula.

L155. Perhaps I misplaced that information, but here the same measurement than in L151 is described?

L199. The realised effective population size estimated from individual increase of inbreeding proposed by Gutiérrez et al. (2009) should not be mixed with the effective population size (Ne; Falconer and Mackay, 1996) estimated from the increase of inbreeding. An clarification of the used measure and the appropriate symbols should be used.

L212. Please, add clarification ‘…where si (OR di)… among all sires (OR dams)…’

Results.

L242. Table 1. I would recommend, here and all across the manuscript, the follow into results (text and tables) the same order as the described in methods section. If in L231 and table title fe, fa and fg are described, the order of the results columns should be the same. Moreover, the table contains results with one, two, or three decimal numbers; please homogenize the number of decimals here and across the manuscript.

L247. Figure 1. Please if as a consequence of the previous comment the order of the columns has been changed, follow the same criteria here. Besides, maybe I am mistaken, but the statistics or the objective of ‘fe/1’ is not described in methods section.

L236-240. Same as in the previous comment (fa, fg order).

L266. Please, rewrite this sentence to clarify (lowest/lower).

L276. Figure 2. It would ease the understanding to keep the same order as in methods and in Table 2, describing firstly lost due to genetic drift and then lost due to unequal contribution. This also affects to text L270-L281.

L290. 0.08 à 0.079 (same as in the table) this is not the only case (L293, L295).

L321. Figure 3. I think that the color of the legend are not right for Ne. Moreover, it would facilitate the understanding to turn to order of breed, so that higher values of Ne are close the axis with descriptive values.

L311. The text is repeated into L298. In addition, the mean value has a decimal more than in table. Homogenize please.

L355. There is probably a typo.

L357. There is probably a typo.

L388. Table 4 and Table 5. Here you have five decimal numbers, are they really necessary?

Discussion.

L487. Please, rewrite this sentence to clarify (was with).

L544. There is probably a typo.

L606. Possibly, here the symbols for ancestral inbreeding and new inbreeding would be more correct to be between brackets.

Author Response

We thank the reviewer for the valuable comments and recommendations to improve our manuscript. We revised our manuscript accordingly.

Reviewer 2:

The paper describes different demographic measures of genetic diversity based on pedigree information to analyze the current scenario on German sheep breeds.  I believe that the topic is interesting and the paper is well structured and written.

Any way I have some comments, in detail:

Abstrac.

L45. It would be worth to mention the Ne range of ‘the next most endangered breeds’

Amended:

Line 46:

The next most endangered breeds are exotic, country and heath breeds with average Ne of 66, 83 and 89, respectively.

L46. Perhaps a sort description of the obtained genetic diversity on other breeds groups (e.i. land sheep, country sheep,…)

Amended: line 46: All groups mentionend now.

Introduction.

L60. Among the treats affecting European breeds, I would mention also the introduction of other foreign and more productive breeds that endangered autochthonous breeds.

Amended:

Lines 59-62:

Intensive production, improvement of autochthonous breeds with other foreign and more productive breeds and increased commercial demands, particularly since the last 50-70 years have contributed significantly to the threats facing European sheep breeds [3].

L63. When the ‘livestock sector’ is mentioned, the whole sector is considered or the sheep sector?

All livestock including sheep sector.

Amended:

Line 65:

especially in the livestock sector including the sheep sector

L84. Please clarify if the described categories depending on breeding directions are fixed, there is no double purpose breeds?

Line 83-85

Amended:

Lines 86-89:

They are grouped into economic and land breeds and can be further differentiated into meat, merino, milk, country and hair sheep depending on their main breeding or a specific objective with only one exception. Merinos are also bred for meat production.

L91. May there be a typo, it seem like it should not be a full stop.

Amended:

Line 91-94:

Breeding for specific traits enforces unequal founder contributions and loss of founder genomes, leading to a reduction in genetic variability, as shown in the Swedish Gute [2], Nellore [6], Valachian [7], Xalda [13], Santa Inês [14], Zandi [15], and Afshari [16] breeds.

L101. I would rewrite the sentence as ‘autochthonous breeds of all breeding directions of German are represented’ to clarify that the considered breeding directions correspond to the autochthonous breeds.

Amended as proposed:

Lines 108-110:

Among the breeds studied, autochthonous breeds of all breeding directions (merino, meat, country, milk) in Germany are represented.

M&M.

This point contains a lot of measures being long and messy, what makes difficult the understanding. Please consider adding subtitles or any other organizer to clarify and ease.

Lines 106-219:

Amended: lines 231-402: completely revised.

  1. Materials and Methods

For our analysis, we used pedigree data of 1,435,562 sheep from 35 different sheep breeds. The reference population were sheep born between 2010 and 2020 and included 981,093 animals (Table 1). Pedigree data was extracted from serv.it OviCap by Vit/Verden, Germany. The OviCap database has been managed by vit/Verden since 2007 and has been available to all sheep breeders since 2013 as an internet-based platform for viewing performance data, as well as for their own work with data from the herdbook and national evaluations of breeding values. All pedigree entries are checked for consistency during data entry and errors must be corrected before the data are accepted. Breeders can use pedigree data from OviCap to avoid inbreeding in the next generation of lambs when planning future matings. In addition, breeders can obtain information on male founders and their contribution to the animals of the current and next generation.

Estimates of demographic measures of genetic diversity were obtained for the respective reference populations of each breed using the PEDIG software [19]. Data editing and calculation of individual rates of inbreeding and realized effective population sizes were done using SAS, version 9.4 (Statistical Analysis System, Cary, NC, USA, 2022).

In brief, we calculated the number of equivalent complete generations (GE) to determine the degree of completeness of the pedigrees in the total dataset [20] as well as the generation intervals (GI) for sires and dams [21]. The demographic measures of genetic diversity estimated included number of founders (f), number of effective founders (fe), effective number of founder genomes (fg), effective number of ancestors (fa) and the ratios fe/f, fa/fe and fg/fe [22,23] based on following formulas:

 , , and ,

with f = number of founders, a = number of ancestors, qk = probability of gene origin of the individual ancestor (k), rk = expected proportion of founder alleles that have been kept within the descendant population, marginal genetic contribution of an individual ancestor (qj).

In addition, we derived the amount of genetic diversity (GD) resulting from unequal contribution of founders and genetic drift [18,24] as follows:

The amount of genetic diversity (GD*) due to an unequal contribution of founders was:

The loss of genetic diversity resulting from unequal contribution of founders is given by 1 – GD* and from genetic drift by GD* - GD. The relative amounts of these losses of genetic diversity can be calculated as proportions of the sum of 1-GD* and GD*-GD.

We estimated individual inbreeding coefficients (F) according to Meuwissen and Luo [25] using PEDIG [19], the genedrop method (Fgd), ancestral inbreeding coefficients according to Ballou [26] (Fa_Bal), ancestral (Fa_Kal) and new (FNew) inbreeding coefficients according to Kalinowski et al. [27] and ancestral history coefficients (AHC) defined by Baumung et al. [28] using GRAIN, version 2.2 [28, 29]. In the data analysis of the reference population, we distuinguished estimates for F for all animals and only for inbred animals (Finbred).

We calculated the degree of deviation of random mating from Hardy–Weinberg proportions as indicator of genetic substructure by comparing the average coancestry within the parental population (Φ) with the average coefficient of inbreeding according to Meuwissen and Luo [25] in the reference population as follows [30]:

The individual rate of inbreeding (ΔFi) was adjusted to GE according to Gutiérrez et al. [31] and calculated as follows:

The realized effective population size (Ne) was derived from the mean ΔFi () of the reference population [32]:

The unbalanced use of male and female breeding animals is expressed by the effective number of sires (NeffS) and dams (NeffD):

where si (or di) is the relative frequency of use of the sire or dam i among all sires (or dams) of the reference population [33].

The distributions of the demographic parameters fe, fg, fa, and their ratios, losses due to unequal contribution of founders and genetic drift, number and effective number of dams and sires are presented for the 35 breeds using boxplots. The boxplots show, in addition to the median, the lower and upper quartiles, as well as the upper and lower whisker, each marking 1.5 times the interquartile range. In order to differentiate the 35 different breeds, we have chosen bee swarm models for the plots.

We calculated for all different inbreeding coefficients means by breed and birth year to analyse trends in these parameters over time. The model applied included breed as a fixed effect and a linear regression coefficient within breed on birth year encoded 1 (=2010) to 11 (=2020). Pearson correlation coefficients among the different coefficients of inbreeding were calculated using SAS. In addition, we calculated Pearson correlation coefficients for individual and ancestral inbreeding coefficients between offspring and parents as well as between both parents using SAS.

L107. Please provide a table showing the number of individuals in each pedigree, reference population and any other interesting descriptive data. Also it would be helpful for the general understanding to specify what the breeding direction of each breed is. Some of these things are included in Table 1, but it is not specified here and the information gets lost over the work.

Amended: we refer here to Table 1, which contains number of individuals in the total dataset, reference dataset and further parameters. Due to the length of the manuscript we did not add a further Table. We added the breeding direction in Table 1.

Amended:

Lines 233:

For our analysis, we used pedigree data of 1,435,562 sheep from 35 different sheep breeds. The reference population were sheep born between 2010 and 2020 and included 981,093 animals (Table 1).

L126-L132. Please make sure that all the elements included into the formulas are described into the text. As far as I can see ‘a’, or ‘j’ are not described in some points.

Amended: see above.

Lines 246-256:

The demographic measures of genetic diversity estimated included number of founders (f), number of effective founders (fe), effective number of founder genomes (fg), effective number of ancestors (fa) and the ratios fe/f, fa/fe and fg/fe [22,23] based on following formulas:

 , , and ,

with f = number of founders, a = number of ancestors, qk = probability of gene origin of the individual ancestor (k), rk = expected proportion of founder alleles that have been kept within the descendant population, marginal genetic contribution of an individual ancestor (qj).

L135. I would specify which is the ‘individual contribution of founder’ that has to be replaced into the formula.

Amended:

See above.

L155. Perhaps I misplaced that information, but here the same measurement than in L151 is described?

Amended:

Deleted: Losses due to genetic drift are calculated as follows: GD* - GD

Section completely revised. See above.

L199. The realised effective population size estimated from individual increase of inbreeding proposed by Gutiérrez et al. (2009) should not be mixed with the effective population size (Ne; Falconer and Mackay, 1996) estimated from the increase of inbreeding. An clarification of the used measure and the appropriate symbols should be used.

Amended:

Lines 282 and everywhere in the text.

The realized effective population size (Ne) was derived from the mean ΔFi () of the reference population [32]:

L212. Please, add clarification ‘…where si (OR di)… among all sires (OR dams)…’

Amended:

Lines 288-289:

where si (or di) is the relative frequency of use of the sire or dam i among all sires (or dams) of the reference population [33].

Results.

L242. Table 1. I would recommend, here and all across the manuscript, the follow into results (text and tables) the same order as the described in methods section. If in L231 and table title fe, fa and fg are described, the order of the results columns should be the same. Moreover, the table contains results with one, two, or three decimal numbers; please homogenize the number of decimals here and across the manuscript.

Amended: the same order of results as in methods.

Lines 248-251:

The demographic measures of genetic diversity estimated included number of founders (f), number of effective founders (fe), effective number of founder genomes (fg), effective number of ancestors (fa) and the ratios fe/f, fa/fe and fg/fe [22,23] based on following formulas:

L247. Figure 1. Please if as a consequence of the previous comment the order of the columns has been changed, follow the same criteria here. Besides, maybe I am mistaken, but the statistics or the objective of ‘fe/1’ is not described in methods section.

Amended: and the ratios fe/f, fa/fe and fg/fe [22,23] based on following formulas:

L236-240. Same as in the previous comment (fa, fg order).

Same order as in the methods.

L266. Please, rewrite this sentence to clarify (lowest/lower).

Amended:

Lines 353-356:

The highest number of ancestors explaining 90% of the population diversity reached 1000 ancestors in German Mutton Merino, but the lowest in Black Mountain with 48 ancestors. The average number of ancestors explaining 30%, 50%, 70% and 90% of the population diversity across all breeds was 8.74, 23.94, 62.77 and 273.89, respectively.

L276. Figure 2. It would ease the understanding to keep the same order as in methods and in Table 2, describing firstly lost due to genetic drift and then lost due to unequal contribution. This also affects to text L270-L281.

Amended: in the same order as methods

Amended:

L290. 0.08 à 0.079 (same as in the table) this is not the only case (L293, L295).

Amended:

Line 382:

with the highest value of 0.079 (WAD)

Line 386:

0.018 (MLW) to 0.100 (IDF).

Line 387:

an average of 0.660.

L321. Figure 3. I think that the color of the legend are not right for Ne. Moreover, it would facilitate the understanding to turn to order of breed, so that higher values of Ne are close the axis with descriptive values.

Amended: Legends for Y-axes changed.

L311. The text is repeated into L298. In addition, the mean value has a decimal more than in table. Homogenize please.

Amended:

Text in Line 298 removed.

L355. There is probably a typo.

Amended:

Line 453:

for α, the lowest individual rate of inbreeding and the largest Ne. In contrast, mountain-

L357. There is probably a typo.

Amended:

Line 455:

from founders, the highest individual rate of inbreeding, the lowest Ne, and the lowest

L388. Table 4 and Table 5. Here you have five decimal numbers, are they really necessary?

Amended: reduced to 3 decimal numbers in Tables 4 and 5.

Discussion.

L487. Please, rewrite this sentence to clarify (was with).

Amended but sentence removed.

In Merinos, the average number of offspring per sire per birth year was 68.4, significantly higher than in all other breeding directions, with estimates ranging from 10.0 to 26.3.

L544. There is probably a typo.

Amended:

Line 663:

In the breed AST, a slightly negative trend for rate of inbreeding was observed

L606. Possibly, here the symbols for ancestral inbreeding and new inbreeding would be more correct to be between brackets.

Amended:

Lines 718-746:

The present study shows the development of demographic measures of genetic diversity and trends of inbreeding for autochthonous sheep breeds and breeds for wool, meat and milk production in Germany. The across-breed analysis revealed losses of genetic diversity mainly due to unequal contributions of founders in the past. This parameter had the largest impact in all 35 breeds on new inbreeding (Fa_New), and new inbreeding was much more important for individual rates of inbreeding and realized effective population sizes compared to ancestral inbreeding and the classical inbreeding coefficient. Trends for individual rates of inbreeding were negative in 11 of the 35 breeds analysed, which may demonstrate the efficiency of maintaining breed diversity. However, the large differences in population structure and breeding history in the different breeds makes it necessary to consider the trends for new and ancestral inbreeding and their impact for breed conservation. In addition, we demonstrated significant differences between the different breeding directions. Mountain-stone sheep breeds had the lowest average Ne value with <50, the lowest number of founders and founder genomes, a relatively high coancestry coefficient on average, positive trends for ancestral inbreeding (Fa_Kal), but negative trends for new inbreeding (Fa_New) and an increasing trend in population sizes and number of effective sires. On the contrary, merino breeds had an average Ne >150 and an expected increase of inbreeding <5% within 50 years, but a negative trend for population sizes and number of effective sires. Annual analyses of demographic data and their trends can be made available through the OviCap national database. This should help to critically review genetic diversity conservation efforts and focus on the most endangered breeds. Along with these analyses, breed associations and breeders can be informed on unequal use of founders and which founder lines are highly threatened and need special consideration in the breeding program. The results of the present study should strengthen the efforts for maintaining the genetic diversity of sheep breeds with a focus on the different ecosystems in Germany.

Reviewer 3 Report

Justinski et al. performed a very interesting, and comprehensive study on a large number of German sheep breeds, using the national vit/Verden database of sheep breeding in Germany. The major focus of this study is investigating the effect of various demographic events (bottleneck, founder events, etc.) underlying the loss of genetic diversity in today’s sheep breeds. The completeness of the pedigree dataset is impressive, making their results strong and reproducible. The authors found the loss of genetic diversity in all 35 breeds studied, and highlighted the effect of past genetic drift as a major underlying factor, rather than past unequal founder contributions. In addition, they measured various inbreeding parameters, and effective population size in a large number of sheep individuals (1.4 M). They found large amount of inbreeding in individuals belong to sheep breeds, intensively bred for meat, as well as mountain sheep, while lower inbreeding was observed in country sheep.

The results are convincing and would be very beneficial for sheep research community, and industrial farming. The results are very well presented through figures, and tables both in the main as well as the supplementary materials. However, the major message of the paper is lost among lines of reports on different values obtained for sheep breeds. I suggest to re-write the discussion, and highlight the major results of this study in relation to the previous one, and how this study fills the current gap. How the findings can actually be helpful for the management as well as breeding programs. How they plan to communicate their results with the German sheep breeding organizations. What are the limitations of their study, and how future studies can build and expand based on this research.

Below, there are few minor comments that I hope that would be helpful and improve the manuscript:

Line 85-87: please add few references to show the cases that intensive breeding let to unequal founder contributions, and consequently loss of genetic diversity.

Line 87-91: that paragraph is basically in the same line as paragraph 85-87, perhaps merge them and use the same references as requested above, unless it meant to cover livestock species beyond sheep.

Line 95: replace “the monitoring” with “monitoring”.

Line 108-109: if appropriate, use either OVICAP , or OviCap uniformly throughout the manuscript.

Line 242: In Table 1, please separate the breeds into two major categories as 1) German autochthonous and 2) intensely selected breeds for various purposes.

Line 352: replace “demografic”, with “ demographic”

Line 412: in Discussion segment there are many sentences that reads like Results – if the results are compared to other studies, please keep them in the discussion, and if not, move them to the result section. At this state, the discussion is too long, and is mainly like repetition of the results. In addition, there is no paragraph, describing how this research is filling the gap in the field, and what are the limitation of this study.

Line 610-612: please be more specific, and describe how these results can be translated to more effective, and evidence-based management plans. The conclusion is also a bit like Results – perhaps just limit this to the major points discovered, limitations, and future directions.

Line 664: replace “Federal” with “The Federal”

Best Wishes,

Author Response

We thank the reviewer for the valuable comments and recommendations to improve our manuscript. We revised our manuscript accordingly.

Reviewer 3:

Justinski et al. performed a very interesting, and comprehensive study on a large number of German sheep breeds, using the national vit/Verden database of sheep breeding in Germany. The major focus of this study is investigating the effect of various demographic events (bottleneck, founder events, etc.) underlying the loss of genetic diversity in today’s sheep breeds. The completeness of the pedigree dataset is impressive, making their results strong and reproducible. The authors found the loss of genetic diversity in all 35 breeds studied, and highlighted the effect of past genetic drift as a major underlying factor, rather than past unequal founder contributions. In addition, they measured various inbreeding parameters, and effective population size in a large number of sheep individuals (1.4 M). They found large amount of inbreeding in individuals belong to sheep breeds, intensively bred for meat, as well as mountain sheep, while lower inbreeding was observed in country sheep.

The results are convincing and would be very beneficial for sheep research community, and industrial farming. The results are very well presented through figures, and tables both in the main as well as the supplementary materials. However, the major message of the paper is lost among lines of reports on different values obtained for sheep breeds. I suggest to re-write the discussion, and highlight the major results of this study in relation to the previous one, and how this study fills the current gap. How the findings can actually be helpful for the management as well as breeding programs. How they plan to communicate their results with the German sheep breeding organizations. What are the limitations of their study, and how future studies can build and expand based on this research.

Below, there are few minor comments that I hope that would be helpful and improve the manuscript:

Line 85-87: please add few references to show the cases that intensive breeding let to unequal founder contributions, and consequently loss of genetic diversity.

Line 87-91: that paragraph is basically in the same line as paragraph 85-87, perhaps merge them and use the same references as requested above, unless it meant to cover livestock species beyond sheep.

Amended:

Lines 91-94:

Breeding for specific traits enforces unequal founder contributions and loss of founder genomes, leading to a reduction in genetic variability, as shown in the Swedish Gute [2], Nellore [6], Valachian [7], Xalda [13], Santa Inês [14], Zandi [15], and Afshari [16] breeds.

Line 95: replace “the monitoring” with “monitoring”.

Amended:

Line 104:

monitoring

Line 108-109: if appropriate, use either OVICAP , or OviCap uniformly throughout the

manuscript.

Amended:

Line 233, 234 and everywhere

we use now everywhere OviCap

Line 242: In Table 1, please separate the breeds into two major categories as 1) German autochthonous and 2) intensely selected breeds for various purposes.

Amended:

Table 1 revised and breeding directions added.

Line 352: replace “demografic”, with “ demographic”

Line 449:

Amended: demographic

Line 412: in Discussion segment there are many sentences that reads like Results – if the results are compared to other studies, please keep them in the discussion, and if not, move them to the result section. At this state, the discussion is too long, and is mainly like repetition of the results. In addition, there is no paragraph, describing how this research is filling the gap in the field, and what are the limitation of this study.

Amended: Repeats of results in the discussion section deleted.

Filling the gap:

This is the first study comprising all sheep breeds in Germany and all data from the different breeding organizations. In the few previous German studies, it was not possible to collate all data from the different breeding organizations nor to have pedigrees as complete as possible as now.

Lines: 515-519:

The aim of this study was to assess genetic diversity and inbreeding trends comprising all autochthonous German sheep breeds and sheep of all breeding directions including wool, meat and milk. There is a great interest in genetic diversity studies in domestic animals with special focus on ruminants [34]. The present study also aims to demonstrate the efficiency of breed conservation programs underway in Germany.

Limitations:

Lines 520-538:

Completeness and depth of pedigree data is crucial for estimation of pedigree-based measures of genetic diversity and inbreeding. All but two autochthonous and all but two imported breeds reached >3.5 equivalent generations. Equivalent generations >4 and >5 had 27/35 and 23/35 breeds, respectively. Similar to the Santa Inês sheep (2.26) [14] and the Kermani sheep (2.22) [35], the breeds Baraka (2.55), Charollais (3.08), and Lacaune (3.09) showed pedigrees with the lowest depth. Summarizing previous reports (Table S9), equivalent generations were 5.42 on average, and 24/33 with >4 and 17/33 with >5 equivalent generations and in the present study with an average of 5.77 equivalent generations in the same range. Because of the importance of pedigree completeness and quality, equivalent generations should be four and larger [36, 37]. Six breeds (BLS, MLS, OMS, SKF, WBS and WHH) reached equivalent generations near 8 or even >8, which is similar to the summary of previous reports with Bleu du Maine [38], Lacaune Confederation, Lacaune Ovitest [39], German Whitehead Mutton [8], Romanov in vivo [38], and Belgium Milk Sheep [40]. Breeds imported a few decades ago like BDC, CHA, LAC, IDF, and SWS do not have deep pedigrees due to their short breeding history in Germany. NOL was developed in recent years as hair sheep in Germany and has, therefore, only a short history as own officially recognized breed. The mountain sheep GBS and SBS are colour variants arisen from WBS and have been recognized as an independent breed with its own herdbook just for a few decades.

Lines 643-653:

In the across-breed comparisons of the present study, loss of genetic diversity due to unequal contribution of founders was found to be the most important factor leading to increasing individual rates of inbreeding and smaller realized effective population sizes. The greatest influence on the loss of genetic diversity due to unequal contributions from founders was the effective number of sires and dams. Data from the across-breed comparisons showed that high effective numbers of dams and high numbers of founders, and high effective numbers of sires and high effective numbers of founders were mutually dependent. Our data suggest that breeding organizations and breeders should be concerned with keeping the number of breeding dams and sires high and using dams and sires as equally as possible in breeding programs.

Line 610-612: please be more specific, and describe how these results can be translated to more effective, and evidence-based management plans. The conclusion is also a bit like Results – perhaps just limit this to the major points discovered, limitations, and future directions.

Amended:

Lines 718-746:

The present study shows the development of demographic measures of genetic diversity and trends of inbreeding for autochthonous sheep breeds and breeds for wool, meat and milk production in Germany. The across-breed analysis revealed losses of genetic diversity mainly due to unequal contributions of founders in the past. This parameter had the largest impact in all 35 breeds on new inbreeding (Fa_New), and new inbreeding was much more important for individual rates of inbreeding and realized effective population sizes compared to ancestral inbreeding and the classical inbreeding coefficient. Trends for individual rates of inbreeding were negative in 11 of the 35 breeds analysed, which may demonstrate the efficiency of maintaining breed diversity. However, the large differences in population structure and breeding history in the different breeds makes it necessary to consider the trends for new and ancestral inbreeding and their impact for breed conservation. In addition, we demonstrated significant differences between the different breeding directions. Mountain-stone sheep breeds had the lowest average Ne value with <50, the lowest number of founders and founder genomes, a relatively high coancestry coefficient on average, positive trends for ancestral inbreeding (Fa_Kal), but negative trends for new inbreeding (Fa_New) and an increasing trend in population sizes and number of effective sires. On the contrary, merino breeds had an average Ne >150 and an expected increase of inbreeding <5% within 50 years, but a negative trend for population sizes and number of effective sires. Annual analyses of demographic data and their trends can be made available through the OviCap national database. This should help to critically review genetic diversity conservation efforts and focus on the most endangered breeds. Along with these analyses, breed associations and breeders can be informed on unequal use of founders and which founder lines are highly threatened and need special consideration in the breeding program. The results of the present study should strengthen the efforts for maintaining the genetic diversity of sheep breeds with a focus on the different ecosystems in Germany.

Line 664: replace “Federal” with “The Federal”

Amended:

Line 798:

The Federal

Round 2

Reviewer 1 Report

Paper may be accepted